# C-Reactive protein as a prognostic indicator in hospitalized patients with COVID-19

Milad Sharifpour[1]⊙*, Srikant Rangaraju[2]⊙, Michael Liu[3], Darwish Alabyad[4], Fadi B. Nahab[2], Christina M. Creel-Bulos[1], Craig S. Jabaley[1], on behalf of the Emory COVID-19 Quality & Clinical Research Collaborative[¶]

1 Department of Anesthesiology and Critical Care, Emory University Hospital, Atlanta, Georgia, United States of America, 2 Department of Neurology, Emory University Hospital, Atlanta, Georgia, United States of America, 3 Emory University School of Medicine, Atlanta, Georgia, United States of America, 4 Morehouse University School of Medicine, Atlanta, Georgia, United States of America

⊙ These authors contributed equally to this work.
¶ Membership of the Emory COVID-19 Quality & Clinical Research Collaborative is provided in the acknowledgements.
* milad.sharifpour@emoryhealthcare.org

**Data Availability Statement:** All relevant data are within the manuscript and its Supporting Information files.

## Abstract

Recent studies have reported that CRP levels are elevated in patients with COVID-19 and may correlate with severity of disease and disease progression. We conducted a retrospective cohort analysis of the medical records of 268 adult patients, who were admitted to one of the six cohorted COVID ICUs across Emory Healthcare System and had at least two CRP values within the first seven days of admission to study the temporal progression of CRP and its association with all-cause in-hospital mortality. The median CRP during hospitalization for the entire cohort was 130 mg/L (IQR 82–191 mg/L), and the median CRP on ICU admission was 169 (IQR 111–234). The hospitalization-wide median CRP was significantly higher amongst the patients who died, compared to those who survived [206 mg/L (157–288 mg/L) vs 114 mg/L (72–160 mg/L), p<0.001]. CRP levels increased in a linear fashion during the first week of hospitalization and peaked on day 5. Compared to patients who died, those who survived had lower peak CRP levels and earlier declines. CRP levels were significantly higher in patients who died compared to those who survived (p<0.001). Our findings support the utility of daily CRP values in hospitalized COVID-19 patients and provide early thresholds during hospitalization that may facilitate risk stratification and prognostication.

## Introduction

The severe acute respiratory syndrome coronavirus-2 (SARS-CoV-2) pandemic has taxed global critical care capacity as manifestations of the resulting coronavirus disease 2019 (COVID-19) may include acute respiratory failure requiring mechanical ventilation. Beyond isolated pulmonary disease, COVID-19 has been associated with significant inflammation leading to neurologic, cardiovascular, coagulation, and other end-organ manifestations. Identifying markers of disease severity may therefore help to identify patients at risk of prolonged intensive care or death.

**Funding:** This study was partly supported by internal funds (Department of Anesthesiology and Critical Care, Emory University Hospital). This study was also supported by the National Center for Advancing Translational Sciences of the National Institutes of Health under Award Number UL1TR002378. The content is solely the responsibility of the authors and does not necessarily represent the official views of the National Institutes of Health. The funders had no role in study design, data collection and analysis, decision to publish, or preparation of the manuscript.

**Competing interests:** No competing interests.

Several retrospective analyses have reported the clinical characteristics and outcomes of hospitalized patients with COVID-19 [1–3]. Derangements in laboratory markers of inflammatory response, including C-reactive protein (CRP), have been identified as predictors of clinical severity and complications [1, 4–7]. CRP is an acute-phase, nonspecific marker of inflammation or infection and has been found to broadly correlate with disease severity and treatment response across a variety of infectious and noninfectious conditions [8]. Elevated CRP levels have been previously reported in severe acute respiratory syndrome, Middle East respiratory syndrome, H1N1 influenza [9–11]. Recent studies have reported that CRP levels are elevated in patients with COVID-19 and may correlate with severity of disease and disease progression [7, 12]. As such, CRP holds promise as a potential prognostic biomarker.

Defining the trends and prognostically-relevant thresholds of CRP in critically ill adults with COVID-19 may facilitate risk stratification, and guide clinical management, and help predict resource utilization. We therefore conducted a retrospective observational study of temporal CRP progression and its association with all cause in-hospital mortality. Intensive care unit (ICU) length of stay (LOS), hospital LOS, and intubation rate were analyzed as secondary outcomes.

## Materials and methods

All patients admitted to one of six cohorted COVID ICUs across three Emory Healthcare acute-care hospitals in Atlanta, GA from March 6, 2020 to May 5, 2020 were eligible for inclusion. We identified all patients who were 18 years or older, with SARS-CoV-2 infection confirmed by molecular testing, with a minimum of two CRP values within 7 days of admission, and for whom primary outcome information was available. In response to early reports, daily CRP measurement was incorporated into local treatment protocols and order sets. The included hospitals measure CRP concentration using Beckman Coulter Synchron system(s) (Beckman and Coulter Life Sciences, Indianapolis, IN). Electronic medical records were reviewed from admission through discharge or until the censor date of June 5, 2020. All data were fully anonymized before they were accessed for analyzing. This study was approved by the Emory University Institutional Review Board (IRB #00000425). Given the retrospective nature of the study, informed consent was waived by the IRB. Reporting adheres to guidelines outlined in the REporting of studies Conducted using Observational Routinely-collected health Data (RECORD) (and transparent reporting of a multivariable prediction model for individual prognosis or diagnosis (TRIPOD) statements.

Basic demographic data including age, sex, race, body mass index (BMI), and comorbid diseases were collected. Length of stay, ICU admission rates, length of ICU stay, time of intubation, duration of mechanical ventilation, and final disposition were also recorded. Data were collected through a combination of validated clinical data warehouse queries, a real-time analytics platform, and manual chart abstraction by the Emory COVID Quality and Clinical Research Collaborative (QCRC). Sources and definitions for each data element examined are available in the Supplement. Within the first 7 days of hospitalization, we determined the peak and slope of CRP change and also calculated the median CRP across the entire hospitalization stay for each patient. These three CRP parameters were used as predictors of our primary endpoint. The primary outcome was all-cause in-hospital mortality. Hospital length of stay, ICU length of stay, and intubation rate were secondary outcomes.

Descriptive statistics were used to analyze the data. Categorical variables were expressed as counts with proportions. Continuous and ordinal variables were expressed as means (standard

deviations, SD) or medians (interquartile range, IQR) based on their distribution. Comparisons between means and medians of continuous variables were assessed with independent (i.e, two-tailed) T-tests or by the Mann-Whitney U test for nonparametric variables, respectively. Categorical variables were compared with Pearson's chi-square and Fisher Exact tests based on sample size. Significance for all descriptive analyses was set at $p<0.05$.

Preliminary univariable analyses were conducted using binary logistic regression for the primary outcome of death to identify potential predictors and included age, race, body mass index, sex, disease severity, ICU admission, intubation status, and comorbid conditions. Statistically significant univariate predictors ($p<0.05$) were then considered in multivariable binary logistic regression analyses to identify independent predictors of the outcome variables (adjusted $p<0.05$). Multicollinearity was assessed by correlation coefficients between two variables and a correlation coefficient of $>0.7$ was considered indicative of multicollinearity, in which case, only one of the two variables was included in the same regression model.

Additionally, we performed receiver operator characteristic (ROC) analyses to determine the discriminative power (area under the curve or AUC) of each CRP parameter for death. All statistical analyses were performed using SPSS statistical package (Version 26), and Graphpad Prism (Version 6.0) were used for graphical representation of data.

## Results and discussion

During the study period, there were a total of 288 unique admissions to a designated COVID-19 ICU, and of those 268 met study inclusion criteria with at least two CRP values within the first seven days of hospitalization. Of those, 203 (75%) had seven daily CRP values within the first seven days of hospitalization. The mean age of the cohort was 63 (15); 119 patients (44.4%) were women, and 184 patients (68.7%) were African American. Hypertension (197 [73.5%]), obesity (141 [52.6%]), diabetes mellitus (118 [44.0%]), and a history of tobacco use (72 [26.8%]) were the most common comorbidities. Demographic and clinical data of the patients are summarized in Table 1.

The median duration of hospitalization was 16 (IQR 10–26) days and the median duration of ICU stay in this cohort was 9 days (IQR 4–16); 200 (74.6%) patients were intubated, and the median duration of mechanical ventilation was 10 (5–14) days.

A median of 11 CRP studies (IQR 7–18) were performed per patient during the entire hospitalization. The median CRP over the total hospitalization period for the entire cohort was 130 mg/L (IQR 82–191 mg/L), and the median CRP on ICU admission was 169 (IQR 111–234). The hospitalization-wide median CRP value was significantly higher amongst the patients who died, compared to those who survived [206 mg/L (157–288 mg/L) vs 114 mg/L (72–160 mg/L), $p<0.001$]. We assessed the trajectory of CRP levels for the entire cohort and observed that CRP levels increased in a linear fashion during the first week of hospitalization peaking on day 5, after which CRP levels decreased continuously (Fig 1). As compared to those who died, patients who survived had lower peak CRP levels and earlier declines in CRP levels (Fig 2A). CRP levels were significantly higher in patients who died compared to those who survived ($p<0.001$) (Fig 2B).

Within the first 7 days, the maximum CRP was significantly higher in patients who died [median 309 mg/L (246–387 mg/L)] compared to those who survived [median 234 mg/L (148–312 mg/L), p = 0.01]. The slope of change in daily CRP levels within the first 7 days was also greater in patients who died [22.6, (5.12–41.7)] compared to those who survived [-0.84, (-18.4–13.4), $p<0.001$]. Receiver operator characteristics (ROC) curve analyses demonstrated median CRP values during the entire hospitalization had a moderate discriminative power to predict mortality (AUC = 0.83). This likely reflects a ceiling for the association between CRP

**Table 1. Patient characteristics and outcomes.**

| Variables | Total cohort (n = 268) | Survived (n = 201) | Died (n = 67) | p-value |
|---|---|---|---|---|
| *Demographics* | | | | |
| Age, Mean (SD) | 63 (15) | 60 (15) | 71 (13) | <0.001 |
| Race AA, n (%) | 184 (68.7%) | 137 (68.2%) | 47 (70.1%) | 0.76 |
| Sex, Female, n (%) | 119 (44.4%) | 94 (46.3%) | 26 (38.8%) | 0.29 |
| Body Mass Index (kg/m2) | 31.9 (8.4) | 33.1 (8.9) | 28.3 (5.6) | <0.001 |
| Smoking (Ever), n (%) | 72 (26.8%) | 48 (23.9%) | 24 (35.8%) | 0.27 |
| Chronic Obstructive Pulmonary Disease, n (%) | 18 (6.7%) | 12 (6%) | 6 (9%) | 0.4 |
| Diabetes Mellitus, n (%) | 118 (44%) | 68 (33.8%) | 21 (31.3%) | 0.23 |
| Hypertension, n (%) | 73.5%) | 142 (70.6%) | 55 (82.1%) | 0.07 |
| Obesity (BMI> = 30kg/m2), n (%) | 141 (52.6%) | 117 (58.2%) | 24 (35.8%) | 0.002 |
| Obstructive Sleep Apnea, n (%) | 23 (8.6%) | 20 (10%) | 3 (4.5%) | 0.21 |
| Coronary Artery Disease, n (%) | 36 (13.4%) | 22 (10.9%) | 14 (20.9%) | 0.04 |
| Stroke, n (%) | 36 (13.4%) | 25 (12.4%) | 11 (16.4%) | 0.41 |
| Congestive Heart Failure, n (%) | 37 (13.8%) | 29 (14.4%) | 8 (11.9%) | 0.61 |
| Chronic Kidney Disease, n (%) | 49 (14.6%) | 27 (13.5%) | 12 (17.9%) | 0.52 |
| *Hospitalization Details* | | | | |
| SOFA (ICU admission) | 6 (4–9) | 6 (3–9) | 7 (4–11) | 0.03 |
| Hosp duration, Median (IQR) | 16 (10–26) | 19 (11–28) | 12 (7–16) | <0.001 |
| Length of ICU stay (Median, IQR) | 9 (4–16) | 9 (3.5–16) | 9 (5–15) | 0.94 |
| Intubation, n (%) | 200 (74.6%) | 141 (70.1%) | 59 (89.8%) | 0.004 |
| Duration of Intubation, Median (IQR) | 10 (5–14) | 10 (5–14) | 11 (5–14) | 0.95 |
| Continuous Renal Replacement Therapy | 62 (23.1%) | 30 (14.9%) | 32 (47.8%) | <0.001 |
| Died | 67 (25%) | | | |
| Peak Troponin ng/mL | 0.04 (0.03–0.09) | 0.03 (0.03–0.07) | 0.06 (0.03–0.19) | <0.001 |
| WBC (on ICU admission) 10E3/mcL | 8.6 (6.2–11.5) | 8.5 (6.2–11.4) | 9.4 (5.7–12.1) | 0.78 |
| D-dimer (on ICU admission) ng/mL | 1475 (870–3982) | 1442 (869–3619) | 1518 (1017–5995) | 0.14 |
| CRP (on ICU admission) mg/L | 169 (111–234) | 168 (108–227) | 173 (119–255) | 0.17 |
| Median CRP (hosp wide) | 129.6 (82.4–191.4) | 114.1 (72–160.4) | 206.3 (157.1–287.8) | <0.001 |
| Max CRP (d1-7) | 247.5 (172.4–340.6) | 234 (148–312) | 308.9 (245.6–386.9) | <0.001 |
| Slope CRP (d1-7) | 4.68 (-13.1–19.8) | -0.84 (-18.37–13.4) | 22.56 (5.12–41.7) | <0.001 |

levels and mortality. In comparison to hospitalization-wide median CRP, we found a nearly equivalent predictive association between the slope of change in CRP level during the first 7 days (rate of change of CRP per day) of hospitalization and mortality (AUC = 0.77). In contrast, we observed a much weaker predictive association between the maximum CRP value during the first 7 days of hospitalization and mortality (AUC = 0.69) (Fig 2C).

In univariate analysis, age (OR 1.057, 95% CI 1.033–1.081, p<0.001), BMI (OR 0.915, 95% CI 0.875–0.958), history of CAD (OR 2.149, 95% CI 1.029–4.49, p = 0.042), CRRT use (OR 5.21, 95% CI 2.813–9.656), intubation (OR 3.138, 95% CI 1.413–6.969, p = 0.005), SOFA score (OR 1.084, 95% CI 1.006–1.164, p = 0.033), median CRP level (OR 1.018, 95% CI 1.013–1.023, p = 0.004), maximum CRP value during the first 7 days (OR 2.68, 95% CI 1.499–4.793, p = 0.001), and the slope of CRP change during the first 7 days (OR 1.044, 95% CI 1.029–1.059, p<0.01), were all predictors of mortality (S1 Table). After adjusting for age, history CAD, CRRT initiation, and the admission SOFA score, the slope of CRP change during the first 7 days remained an independent predictor of mortality (OR 1.03 per unit change, 95% CI 1.014–1.046, p<0.001). We did not observe collinearity between any of the CRP measures with SOFA score (correlation coefficients <0.2). A cutoff value of 10 units/day (rate of change of

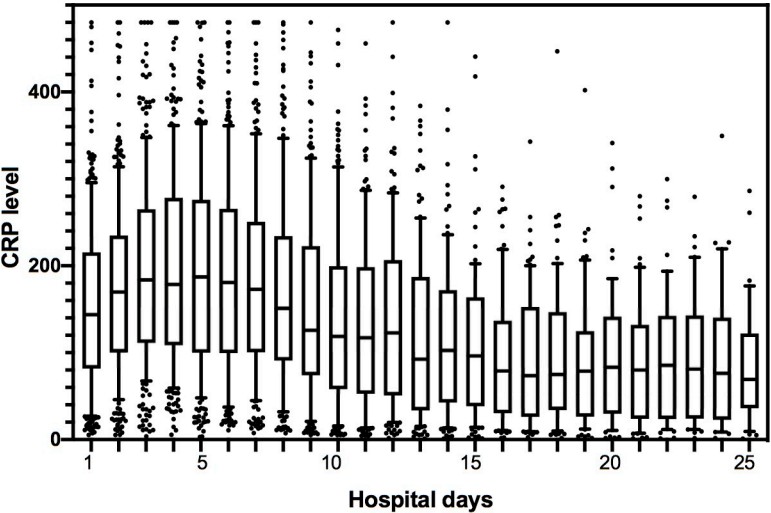

**Fig 1. Trends of change in CRP levels in hospitalized COVID-19 patients (n = 268).** Box plots represent median CRP (with IQR, 10th and 90th percentile and outliers) for the cohort on each given day, up to day 25 of hospitalization.

CRP value) provided 70% sensitivity, 69% specificity, 43% positive predictive value (PPV), and 87% negative predictive value (NPV) for mortality, while a slope of 20 units/day provided 52% sensitivity, 85% specificity, 53% PPV, and 84% NPV for mortality. A maximum CRP during

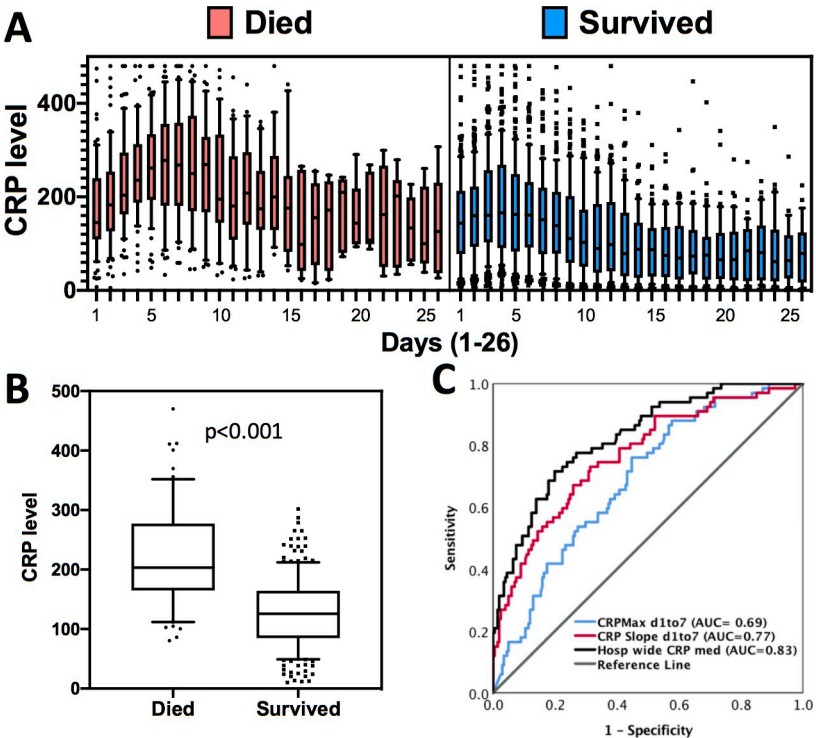

**Fig 2. Association between CRP levels in-hospital mortality in COVID-19 patients.** (A) Trends in CRP changes over time in patients who died (red) and survived (blue). (B) Comparison of median hospitalization-wide CRP levels between patients who died and survived. (C) ROC curve analysis showing predictive association between hospitalization-wide median CRP, maximum CRP within the first 7 days, and the rate of change in CRP (slope) within the first 7 days with mortality. AUC values are indicated.

the first 7 days greater than 250 mg/L provided 67% sensitivity, 57% specificity, 34% PPV, and 84% NPV for mortality. These thresholds for CRP measures are shown in Table 2. Patients with a slope of CRP change during the first 7 days of greater than 10 units/day had a significantly higher odds of death in the unadjusted analysis (OR 5.66, 95% 3.1–10.51, p<0.001). A maximum CRP value during the first 7 days greater than 250 mg/L was associated with a higher odds of death in unadjusted analysis [OR 2.68 (1.5–4.8, p = 0.001)].

In a multivariable analysis adjusting for age, need for CRRT, obesity, SOFA score, intubation status, history of CAD or hypertension, both the slope of CRP rise greater than 10 (OR 2.98, 95% CI 1.44–6.18, p = 0.003) and the maximum CRP value within the first 7 days (OR 2.22, 95% CI 1.00–4.95, p = 0.051) were identified as independent predictors of mortality (Table 3). To ensure lack of collinearity between slope of CRP change, and maximum CRP level within the first 7 days, we confirmed low-level although significant correlation between these two variables (Spearman's rho = 0.18, p = 0.004). Inclusion of CRP slope and maximum CRP in the first 7 days in the final model resulted in a better predictive model (-2log likelihood ratio improvement from 241 to 179 after including CRP variables) as compared to a model without these two variables.

Maximum CRP value, but not the slope of CRP change within the first 7 days, was significantly associated with length of ICU stay (Max CRP Spearman's rho = 0.033 p<0.001; CRP slope Spearman's rho = 003, p = 0.65), as well as with total hospital length of stay (Max CRP Spearman's rho = 0.15 p = 0.015; CRP slope Spearman's rho = -0.09, p = 0.13).

Our analysis of temporal trends in CRP values demonstrated that the median CRP value during the entire hospitalization as well as the rate of CRP change during the first seven days of hospitalization were not only independent predictors of mortality but also correlated with the length of ICU stay in hospitalized COVID-19 patients. In particular, we identified a rate of change of ≥ 20 units per day in the CRP value during the first seven days of hospitalization was associated with significantly increased odds of mortality.

C-reactive protein is a nonspecific, acute-phase, inflammatory protein, whose expression is increased in response to tissue injury, inflammation, and infection [8, 13]. It is widely available, easy to obtain, and inexpensive. Baseline CRP values are affected by factors such as age, sex, lipid profile, and smoking status [14]. CRP values rise within 24 to 72 hours of exposure to noxious stimuli, and decrease exponentially within 18–20 hours of the resolution of the stimuli

**Table 2. Optimal thresholds of maximum and slope of CRP in the first 7 days of hospitalization as predictors of mortality.**

| CRP Max d1to7 | Sensitivity | Specificity | PPV | NPV |
|---|---|---|---|---|
| 150 | 0.955 | 0.259 | 0.3 | 0.95 |
| 200 | 0.881 | 0.393 | 0.4 | 0.94 |
| 250 | 0.672 | 0.567 | 0.34 | 0.84 |
| 300 | 0.537 | 0.726 | 0.4 | 0.83 |
| 350 | 0.418 | 0.826 | 0.45 | 0.81 |
| CRP Slope d1-7 | Sensitivity | Specificity | PPV | NPV |
| 0 | 0.806 | 0.517 | 0.36 | 0.89 |
| 5 | 0.761 | 0.592 | 0.38 | 0.88 |
| 10 | 0.701 | 0.692 | 0.43 | 0.87 |
| 15 | 0.612 | 0.761 | 0.46 | 0.86 |
| 20 | 0.522 | 0.846 | 0.53 | 0.84 |
| 25 | 0.463 | 0.876 | 0.56 | 0.83 |
| 30 | 0.373 | 0.915 | 0.59 | 0.81 |

**Table 3. Multivariable models to identify predictors of mortality in hospitalized COVI19 patients.**

| Model 1[*] | Variable | OR | 95% CI | p |
|---|---|---|---|---|
| | Age | 1.069 | 1.032–1.108 | <0.001 |
| | CRP Slope d1to7 | 1.03 | 1.014–1.046 | <0.001 |
| | CRP tests (count d1 to 7) | 0.55 | 0.386–0.785 | 0.001 |
| | CRRT | 4.379 | 1.844–10.395 | 0.001 |
| | CRP Max d1to7 | 1.007 | 1.003–1.011 | 0.001 |
| | Obesity (BMI> = 30kg/m2) | 0.341 | 0.147–0.792 | 0.012 |
| | Intubation | 4.435 | 1.007–19.527 | 0.049 |
| | SOFA score | 1.038 | 0.93–1.159 | 0.506 |
| | CAD | 1.381 | 0.498–3.826 | 0.535 |
| | Hypertension | 1.142 | 0.44–2.964 | 0.785 |
| Model 2[#] | Variable | OR | 95% CI | p |
| | Age | 1.066 | 1.032–1.102 | <0.001 |
| | CRRT | 4.994 | 2.17–11.491 | <0.001 |
| | CRP tests (count d1 to 7) | 0.621 | 0.464–0.83 | 0.001 |
| | CRP Slope d1to7≥10 | 2.98 | 1.437–6.181 | 0.003 |
| | Intubation | 4.237 | 1.176–15.258 | 0.027 |
| | Obesity (BMI≥30kg/m2) | 0.437 | 0.2–0.956 | 0.038 |
| | MaxCRP d1to7 ≥250 | 2.221 | 0.997–4.948 | 0.051 |
| | CAD | 1.6 | 0.625–4.1 | 0.327 |
| | Hypertension | 1.13 | 0.455–2.809 | 0.792 |
| | SOFA score | 1.004 | 0.905–1.114 | 0.938 |

[*]Model 1: CRP slope and Maximum CRP levels during the first 7 days were used as continuous variables in the model.

[#]Model 2: CRP slope≥0 and Maximum CRP level≥250 were used as thresholds in the model.

[15]. In patients with sepsis, elevated CRP level > 100 mg/L has been identified as an independent predictor of ICU and 30-day mortality as well as length of ICU stay in patients who fulfill Sepsis-3 criteria [16]. Elevated CRP values have been reported in viral respiratory illnesses such as SARS, MERS-CoV, and H1N1, and have been reported to correlate with disease severity and predictors of disease progression [9–11]. Similarly, CRP levels have been reported to be elevated in hospitalized patients with COVID-19, and to correlate with severity of disease and mortality. In one retrospective study, CRP and IL-6 values on admission were independent predictors of disease severity, while in a small prospective study IL-6 and CRP levels correlated with the development of respiratory failure [12, 17]. Another small retrospective study reported a correlation between mortality and CRP values in diabetic patients with COVID-19 [18].

Our results confirm that median CRP value correlates with severity of COVID-19 and is an independent predictor of mortality. Furthermore, our results show that the rate at which CRP level rises during the first seven days of hospitalization could be used as a tool to predict disease progression and the need for early ICU transfer. As compared to the rate of change in CRP, the maximum CRP value within the first seven days was less predictive of death highlighting the potential clinical utility of frequent CRP measurements especially during the early hospital course. Since CRP has been shown to be associated with disease severity in COVID-19, we suspect that the rate of rise in CRP is indicative of disease worsening and a measure of underlying systemic inflammatory responses. Emerging evidence now indicates that inhibition of systemic inflammation by dexamethasone may reduce COVID-19-related mortality, providing evidence for a causal link between systemic inflammation and clinical outcome in COVID-19

patients [19]. Whether the CRP thresholds we have identified can be used to risk-stratify patients early and guide intensive management with respiratory support and/or immunosuppression with corticosteroids, remains to be determined in future prospective studies. Similarly, CRP levels, along with those of other inflammatory markers, can be used to monitor response to therapy [20].

The single center nature of the study limits the generalizability of our findings. However, the relatively large sample size of our study compared to the previous studies and ethnic diversity of our patient population is a strength. Additionally, our study has inherent limitations associated with its retrospective analysis and not all patients had daily CRP values resulting in an inability to evaluate the utility of admission CRP value as a predictor of ICU admission. Some patients were still alive in the hospital at the time of analysis. Future studies using larger datasets and more universal CRP measurements will enable validation of our findings. Including something about selection bias here just to be complete, although it does not seem like a particularly worrisome issue.

## Conclusion

In conclusion, CRP is widely available, inexpensive, and an easy to obtain marker that correlates with disease severity and mortality. Our findings support the utility of daily CRP values in hospitalized COVID-19 patients and provide early thresholds during hospitalization that may facilitate risk stratification and prognostication.

## Supporting information

**S1 Table. Univariate analyses to identify predictors of mortality.**
(TIF)

## Acknowledgments

We would like to thank Emory COVID-19 Quality and Clinical Research Collaborative Members (in alphabetical order): Max W. Adelman, Scott Arno, Sara C. Auld, Theresa Barnes, William Bender, James M. Blum, Gaurav Budhrani, Stephanie Busby, Laurence Busse, Mark Caridi-Scheible, David Carpenter, Nikulkumar Chaudhari, Craig M. Coopersmith, Gordon Dale, Lisa Daniels, Johnathan A. Edwards, Jane Fazio, Babar Fiza, Eliana Gonzalez, Ria Gripaldo, Charles Grodzin, Robert Groff, Alfonso C. Hernandez-Romieu, Max Hockstein, Dan Hunt, Craig S. Jabaley, Jesse T. Jacob, Colleen Kraft, Greg S. Martin, Samer Melham, Nirja Mehta, Chelsea Modlin, David J. Murphy, Jung Park, Deepa Patel, Cindy Powell, Amit Prabhaker, Jeeyon Rim, Ramzy Rimawi, Chad Robichaux, Nicholas Scanlon, Milad Sharifpour, Bashar Staitieh, Michael Sterling, Jonathan Suarez, Colin Swenson, Nancy Thakkar, Alexander Truong, Hima Veeramachaneni, Alvaro Velasquez, Aimee Vester, Michael Waldmann, Max Weinmann, Thanushi Wynn, and Joel Zivot.

Dr. Sara Auld (sara.auld@emory.edu) is the lead author of the Emory COVID-19 Quality and Clinical Research Collaborative.

## Author Contributions

**Conceptualization:** Milad Sharifpour, Srikant Rangaraju, Fadi B. Nahab, Christina M. Creel-Bulos, Craig S. Jabaley.

**Data curation:** Michael Liu, Darwish Alabyad.

**Formal analysis:** Srikant Rangaraju, Michael Liu, Darwish Alabyad.

**Writing – original draft:** Milad Sharifpour.

**Writing – review & editing:** Srikant Rangaraju, Michael Liu, Darwish Alabyad, Fadi B. Nahab, Christina M. Creel-Bulos, Craig S. Jabaley.

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
