## [Decision Letter · Decision Letter 0]

28 Sep 2020

PONE-D-20-25813

C-Reactive Protein as a Prognostic Indicator in Hospitalized Patients with COVID-19

PLOS ONE

Dear Dr. Sharifpour,

Thank you for submitting your manuscript to PLOS ONE. After careful consideration, we feel that it has merit but does not fully meet PLOS ONE’s publication criteria as it currently stands. Therefore, we invite you to submit a revised version of the manuscript that addresses the points raised during the review process.

Some minor edits are needed.

We look forward to receiving your revised manuscript.

Kind regards,

Stelios Loukides

Academic Editor

PLOS ONE

Journal Requirements:

2. In your ethics statement in the Methods section and in the online submission form, please provide additional information about the data used in your retrospective study. Specifically, please ensure that you have discussed whether all data were fully anonymized before you accessed them and/or whether the IRB or ethics committee waived the requirement for informed consent. If patients provided informed written consent to have data from their medical records used in research, please include this information.

3. Please provide the names of the six cohorted COVID ICUs across three Emory Healthcare acute care hospitals in Atlanta, GA.

"No"

5. One of the noted authors is a group or consortium [Emory COVID-19 Quality and Clinical Research Collaborative]. In addition to naming the author group, please list the individual authors and affiliations within this group in the acknowledgments section of your manuscript. Please also indicate clearly a lead author for this group along with a contact email address.

6. Please amend either the abstract on the online submission form (via Edit Submission) or the abstract in the manuscript so that they are identical.

7. We note you have included a table to which you do not refer in the text of your manuscript. Please ensure that you refer to Table 3 in your text; if accepted, production will need this reference to link the reader to the Table.

Reviewers' comments:

Reviewer's Responses to Questions

**Comments to the Author**

1. Is the manuscript technically sound, and do the data support the conclusions?

Reviewer #1: Yes

2. Has the statistical analysis been performed appropriately and rigorously? 

Reviewer #1: Yes

3. Have the authors made all data underlying the findings in their manuscript fully available?

Reviewer #1: Yes

4. Is the manuscript presented in an intelligible fashion and written in standard English?

Reviewer #1: Yes

5. Review Comments to the Author

Reviewer #1: In this manuscript, the authors report a retrospective study investigating the temporal progression of C-reactive protein (CRP) and its association with all-cause hospital mortality. Comparison of patients that died with those that survived showed that hospitalization-wide median CRP levels were significantly higher in patients that died. CRP levels tended to increase linearly during the first week of hospitalization, peaking on day 5. However, patients that survived had lower peak CRP levels and earlier declines compared to patients that died. The authors suggest the utility of daily CRP values in the stratification and prognosis of patients with COVID-19

This is a thorough and interesting study investigating the potential utility of CRP values during hospitalization in assessing disease severity and prognosis in COVID-19 patients. The experimental design of the study is solid and the results and clearly presented. The manuscript is well-written and makes a positive contribution to the field. There are only a few minor revisions recommended.

1. In Table 1, please indicate the corresponding units for the variables (e.g., CRP, troponin, D-dimer, WBC). Please indicate the meaning of the acronyms for the clinical variables as well (e.g., DM, OSA, CAD, CRRT, CHF, CKD).

2. Table 2 shows two different models. Please indicate in the text what is the difference between them.

3. Please indicate in Materials and Methods what was the methodology used to measure CRP levels and whether it was the same for all six hospitals.

4. On lines 119-120, the word “one” is repeated. Did the authrors meant “only one”?

6. PLOS authors have the option to publish the peer review history of their article (what does this mean?). If published, this will include your full peer review and any attached files.

Reviewer #1: No

---

## [Author Response · Author response to Decision Letter 0]

30 Oct 2020

Thank you! I have gone through the manuscript and to the best of my ability followed the PLOS ONE’s style requirements.

2. In your ethics statement in the Methods section and in the online submission form, please provide additional information about the data used in your retrospective study. Specifically, please ensure that you have discussed whether all data were fully anonymized before you accessed them and/or whether the IRB or ethics committee waived the requirement for informed consent. If patients provided informed written consent to have data from their medical records used in research, please include this information.

Thank you. The data were fully anonymized before we accessed them. Since this was a retrospective analysis, the IRB waived the requirement for informed consent. 

3. Please provide the names of the six cohorted COVID ICUs across three Emory Healthcare acute care hospitals in Atlanta, GA.

1. 2 East ICU at Emory St. Joseph Hospital

2. Medical ICU at Emory University Hospital Midtown

3. Peachtree ICU at Emory University Hospital Midtown

4. 5G ICU at Emory University Hospital

5. 6G ICU at Emory University Hospital

6. 5E ICU at Emory University Hospital

"No"

a. Please clarify the sources of funding (financial or material support) for your study. List the grants or organizations that supported your study, including funding received from your institution.

a. This study was partly supported by internal funds (Department of Anesthesiology and Critical Care, Emory University Hospital)

b. This study was also supported by the National Center for Advancing Translational Sciences of the National Institutes of Health under Award Number UL1TR002378.

The content is solely the responsibility of the authors and does not necessarily represent the official views of the National Institutes of Health

a. None of the authors received a salary from the funder.

d. If you did not receive any funding for this study, please state: “The authors received no specific funding for this work.” 

a. Thank you very much. This is not applicable, and we have indicated our source of funding appropriately

5. One of the noted authors is a group or consortium [Emory COVID-19 Quality and Clinical Research Collaborative]. In addition to naming the author group, please list the individual authors and affiliations within this group in the acknowledgments section of your manuscript. Please also indicate clearly a lead author for this group along with a contact email address.

Emory COVID-19 Quality and Clinical Research Collaborative Members (in alphabetical order): Max W. Adelman, Scott Arno, Sara C. Auld, Theresa Barnes, William Bender, James M. Blum, Gaurav Budhrani, Stephanie Busby, Laurence Busse, Mark Caridi-Scheible, David Carpenter, Nikulkumar Chaudhari, Craig M. Coopersmith, Gordon Dale, Lisa Daniels, Johnathan A. Edwards, Jane Fazio, Babar Fiza, Eliana Gonzalez, Ria Gripaldo, Charles Grodzin, Robert Groff, Alfonso C. Hernandez-Romieu, Max Hockstein, Dan Hunt, Craig S. Jabaley, Jesse T. Jacob, Colleen Kraft, Greg S. Martin, Samer Melham, Nirja Mehta, Chelsea Modlin, David J. Murphy, Jung Park, Deepa Patel, Cindy Powell, Amit Prabhaker, Jeeyon Rim, Ramzy Rimawi, Chad Robichaux, Nicholas Scanlon, Milad Sharifpour, Bashar Staitieh, Michael Sterling, Jonathan Suarez, Colin Swenson, Nancy Thakkar, Alexander Truong, Hima Veeramachaneni, Alvaro Velasquez, Aimee Vester, Michael Waldmann, Max Weinmann, Thanushi Wynn, and Joel Zivot.

Dr. Sara Auld (sara.auld@emory.edu) is the lead author of the collaborative.

6. Please amend either the abstract on the online submission form (via Edit Submission) or the abstract in the manuscript so that they are identical.

 Thank you. I have corrected this and both abstracts are now identical. 

7. We note you have included a table to which you do not refer in the text of your manuscript. Please ensure that you refer to Table 3 in your text; if accepted, production will need this reference to link the reader to the Table.

Thank you. We have included a sentence in the results section (line 187) as follows, referencing Table 2. ‘These thresholds for CRP measures are shown in Table 2. Table 3 was changed to Table 2 in order to keep the correct order of Tables in the body of the text. 

1. In Table 1, please indicate the corresponding units for the variables (e.g., CRP, troponin, D-dimer, WBC). Please indicate the meaning of the acronyms for the clinical variables as well (e.g., DM, OSA, CAD, CRRT, CHF, CKD).

Thank you. These are reflected in the text/table now, as well as here.

CRP: mg/L

Troponin: ng/mL

D-Dimer: ng/mL

WBC: 10E3/mcL

DM: Diabetes Mellitus

CAD: Coronary Artery Disease

OSA: Obstructive Sleep Apnea

CRRT: Continuous Renal Replacement Therapy

CHF: Congestive Heart Failure

CKD: Chronic Kidney Disease

2. Table 2 shows two different models. Please indicate in the text what is the difference between them.

We have changed table 2 to table 3 and included a foot note to Table 3 to clarify the differences between Models 1 and 2 as follows:

Model 1: CRP slope and Maximum CRP levels during the first 7 days were used as continuous variables in the model.

Model 2: CRP slope≥0 and Maximum CRP level ≥ 250 were used as thresholds in the model.

3. Please indicate in Materials and Methods what was the methodology used to measure CRP levels and whether it was the same for all six hospitals.

All six hospitals use the same methodology to measure CRP. CRP reagent is used to measure CRP concentration by a turbidimetric method1,2 In the reaction, c-reactive protein combines with specific antibody to form insoluble antigen-antibody complexes. The Beckman Coulter SYNCHRON System(s) � (Beckman Coulter Life Sciences, Indianapolis, IN) automatically proportions the appropriate sample and reagent volumes into a cuvette. The ratio used is one part sample to 26 parts reagent. The system monitors the change in absorbance at 340 nanometers. This change in absorbance is proportional to the concentration of C-reactive protein in the sample and is used by the System to calculate and express C-reactive protein concentration based upon a single-point adjusted, pre-determined calibration curve

4. On lines 119-120, the word “one” is repeated. Did the authors mean “only one”?

Thank you, I have corrected this and eliminated the redundant “one” and changed it to “only one” in line 119.

 

References:

1. Boyden, A., Button, E., Germerog, D., "Precipitin Testing With Special Reference to the Measurement of Turbidity", J. Immunol., 57:211 (1947). 

2. Hellsing, K., "The Effects of Different Polymers for Enhancement of the Antigen-Antibody Reaction as Measured with Nephelometry", Protides of the Biological Fluids, 23:579 (1973).

---

## [Editor Report · Decision Letter 1]

3 Nov 2020

C-Reactive Protein as a Prognostic Indicator in Hospitalized Patients with COVID-19

PONE-D-20-25813R1

Dear Dr. Sharifpour,

We’re pleased to inform you that your manuscript has been judged scientifically suitable for publication and will be formally accepted for publication once it meets all outstanding technical requirements.

Kind regards,

Stelios Loukides

Academic Editor

PLOS ONE
---

## [Editor Report · Acceptance letter]

10 Nov 2020

PONE-D-20-25813R1 

C-Reactive protein as a prognostic indicator in hospitalized patients with COVID-19 

Dear Dr. Sharifpour:

I'm pleased to inform you that your manuscript has been deemed suitable for publication in PLOS ONE. Congratulations! Your manuscript is now with our production department. 

Kind regards, 

on behalf of

Dr Stelios Loukides 

Academic Editor

PLOS ONE